# Lateral Degassing Method for Disposable Film-Chip Microfluidic Devices

**DOI:** 10.3390/membranes11050316

**Published:** 2021-04-26

**Authors:** Suhee Park, Hyungseok Cho, Junhyeong Kim, Ki-Ho Han

**Affiliations:** Center for Nano Manufacturing, Department of Nanoscience and Engineering, Inje University, Gimhae 50834, Gyongnam, Korea; sue020604@naver.com (S.P.); elshaddai88@naver.com (H.C.); hellowjuns@naver.com (J.K.)

**Keywords:** disposable microfluidic device, film-chip technique, lateral degassing method

## Abstract

It is critical to develop a fast and simple method to remove air bubbles inside microchannels for automated, reliable, and reproducible microfluidic devices. As an active degassing method, this study introduces a lateral degassing method that can be easily implemented in disposable film-chip microfluidic devices. This method uses a disposable film-chip microchannel superstrate and a reusable substrate, which can be assembled and disassembled simply by vacuum pressure. The disposable microchannel superstrate is readily fabricated by bonding a microstructured polydimethylsiloxane replica and a silicone-coated release polymeric thin film. The reusable substrate can be a plate that has no function or is equipped with the ability to actively manipulate and sense substances in the microchannel by an elaborately patterned energy field. The degassing rate of the lateral degassing method and the maximum available pressure in the microchannel equipped with lateral degassing were evaluated. The usefulness of this method was demonstrated using complex structured microfluidic devices, such as a meandering microchannel, a microvortex, a gradient micromixer, and a herringbone micromixer, which often suffer from bubble formation. In conclusion, as an easy-to-implement and easy-to-use technique, the lateral degassing method will be a key technique to address the bubble formation problem of microfluidic devices.

## 1. Introduction

The removal of bubbles inside microchannels is the most common challenge for the automation and commercialization of microfluidic devices [1]. Bubbles are often introduced during a microfluidic setup [2] or induced by microscopic air pockets [3] and dissolved gases [4] in microchannels. Once formed, they usually become attached to complex microstructures and microgrooves with a high surface-to-volume ratio [5] and are very difficult to remove. Bubbles can lead to flow instability, thereby degrading the performance of microfluidic devices. They can also block microchannels and interact with sensing surfaces, causing malfunctions. For example, bubbles can considerably increase flow-induced shear stresses, which leads to cell damage and detachment in microfluidic cell cultures [6]. During polymerase chain reactions (PCRs), bubbles damage nucleic acids and cause the PCR solution to overflow in the reaction chamber [7,8]. In drug screening using microfluidic devices, bubbles can change the cellular environment of individual culture chambers by partially interfering with drug diffusion [9]. For this reason, developing a degassing technique that can easily be implemented in microfluidic devices is essential for improving the reliability and reproducibility of microfluidic experiments.

Commercially available degassing systems can remove air bubbles entering microchannels but are not effective at removing air bubbles formed in the microchannels. Another common way to prevent bubble formation is to pre-fill the microchannels with a solution diluted with soft surfactants [10], such as sodium dodecyl sulfate (SDS) and Triton-X, or low-polarity solvents [11,12], such as isopropanol, ethanol, and methanol, during the experimental setup. Even with low-polarity reagents, small bubbles still tend to get trapped on surfaces with a high surface-to-volume ratio. Hydrophilic surface treatment of microchannels with oxygen plasma is effective at preventing air bubbles from adhering to the microchannels [13,14]. The self-priming method using polydimethylsiloxane (PDMS) microfluidic devices [15,16,17,18] can also be used to remove residual air bubbles inside microchannels. However, the oxidized hydrophilicity and the self-priming effect of PDMS are temporary and gradually disappear within a few minutes [19]. Moreover, after air bubbles have formed in the microchannel, all of the methods described above are no longer effective. The on-chip bubble trap structure [6,20] is an effective way to prevent air bubbles from entering the microchannel. However, air bubbles tend to merge and accumulate, so when the air bubbles continue to flow in and become larger than what the trap structure can handle, they eventually invade into microchannels.

To overcome the limitations of these passive anti-bubble techniques, active degassing methods [21,22,23] have been developed, in which air bubbles are usually trapped in a degassing microchamber and simultaneously extracted through a gas-permeable membrane. The gas-permeable membrane consists of a part of the degassing microchamber and is generally heterogeneous from the main material of the microfluidic devices, making their fabrication difficult and less reproducible [24,25,26]. Meanwhile, PDMS is a commonly used material for microfluidic device fabrication [27] and is also known for its high gas permeability [28,29]. Therefore, a variety of active degassing methods using PDMS membranes have been developed for a wide range of microfluidic applications, such as PCR [30], cell culture [31], drug screening [32], and fuel cells [33]. Most PDMS-based active degassing methods use vacuum microchambers for bubble extraction [34,35], which is usually implemented in a multi-layered structure, making device fabrication difficult and impractical. As a simple alternative, in-plane degassing methods has been developed for PDMS-based microfluidic devices [36,37,38,39]. In this case, it is necessary to place degassing lines near the microchannels where air bubbles form frequently. With more advanced microfluidic devices, many degassing lines are required and each line must be connected to vacuum pressure through a tube [40], making the microfluidic setup cumbersome. Furthermore, as the microchannel network becomes more complex, it may not be possible to place the degassing line close to the desired microchannel.

This study introduces an easy-to-use lateral degassing method, which is simple to implement even in complex microfluidic devices. The proposed lateral degassing method does not require cumbersome multi-tubing connections to vacuum pressure, and above all, the degassing lines can be placed close together regardless of where the microchannels that require degassing are placed in the device. The method is implemented using a disposable microchannel superstrate and a reusable substrate, which can be simply assembled and disassembled by applying vacuum pressure (Figure 1a). The disposable superstrate is fabricated by bonding a microstructured PDMS replica with a silicone-coated release polyethylene terephthalate (PET) thin film. The degassing speed and the maximum internal pressure that the microchannel can withstand are evaluated according to different thicknesses of the PDMS degassing wall. To demonstrate its usefulness, the proposed lateral degassing method is applied to complex microfluidic devices that are often severely affected by bubble interference.

## 2. Methods

### 2.1. Design and Working Principle

The disposable microchannel superstrate was fabricated by bonding a 12 μm thick PET film with a microstructured PDMS replica, containing a microchannel network, degassing lines, and a vacuum trench. The PET film forms the bottom layer of the microchannel, and microholes are formed under each degassing line by micro-punching. The vacuum trench is used for vacuum assembly with the reusable substrate and is formed along the edges of the superstrate to enclose the microchannel network and the degassing lines. It also serves as a conduit to exhaust air bubbles in the microchannel through the degassing lines, which must be placed as close as possible to the desired microchannel to improve the degassing speed. The width of the degassing line was designed to be 40 to 100 μm so that it can be applied to complex microchannels without affecting the size of microfluidic devices. The reusable substrate supports the disposable superstrate during vacuum assembly and prevents the PET film from warping and twisting. In addition, to implement various microfluidic functions along with the lateral degassing method, the reusable substrate can be made capable of generating energy fields that can be transmitted through the PET film and can actively manipulate and detect the substances in the microchannel [41,42,43]. The disposable superstrate and the reusable substrate are tightly assembled by vacuum pressure exerted through the vacuum hole. When air bubbles form in the microchannel during vacuum assembly, they are first released through the gas-permeable degassing wall. Eventually, the air bubbles flow along the degassing lines, the microholes, the gaps between the PET film and the substrate, and the vacuum trench before exiting through the vacuum hole (Figure 1b). Air bubbles are extracted due to the gas permeability of the PDMS degassing wall, while the solution remains in the microchannel due to the strong hydrophobicity of PDMS. After use, the vacuum pressure is released to allow the superstrate and substrate to be disassembled, and then the superstrate is replaced for the next operation. Due to its simplicity and high compatibility, the lateral degassing method can be used in a wide range of microfluidic applications.

### 2.2. Fabrication Process

The disposable microchannel superstrate was fabricated by bonding a silicone-coated release PET film (12 µm thick) with a microstructured PDMS replica produced by soft lithography. The microchannel network and the degassing lines (50 µm thick) were patterned using a thick negative photoresist (SU-8 3050, Kayaku Advanced Materials, Westborough, MA, USA) on a slide glass (1.7 mm thick), along with an evaporated 1000 Å Cr layer to increase the adhesion between the SU-8 and the slide glass. A vacuum trench was formed by attaching acrylic bars, with a cross section of 2 × 2 mm^2^, which employs ultraviolet (UV) glue and surrounds the microchannel and the degassing lines (Figure 2a). Liquid-phase PDMS prepared by mixing resin and a curing agent (Sylgard 184, Dow Corning, Midland, MI, USA) at a 10:1 ratio, was poured into the SU-8 mold and cured for 1 h at 75 °C in an oven (Figure 2b). The PDMS replica was peeled off from the SU-8 mold, and the inlet/outlet reservoirs as well as the vacuum hole were formed using a 1.5 mm diameter biopsy punch (Kai Industries, Tokyo, Japan). The silicone-coated release PET film was bonded together with the PDMS replica by treating the oxygen plasma for 120 s at a power of 6.8 W (PDC-32G-2, Harrick Plasma, Ithaca, NY, USA) and curing for 1 h at 75 °C in an oven (Figure 2c). Then, the PET film just below each degassing line was perforated using a probe tip (M1.5ST, MSTECH, Hwaseong, Korea) mounted on a micromanipulator (PS100, MSTECH) to create microholes (Figure 2d). In this study, a bare glass slide was used for the substrate. For the experimental setup, the fabricated disposable microchannel superstrate and the glass substrate were assembled by applying a vacuum pressure (−50 kPa) to the vacuum trench through the vacuum hole (Figure 2e). The applied vacuum strongly attaches the PET film to the substrate to minimize the gap between them. However, as both these components are made of rigid materials, tiny gaps exist between them, as evident in the fringe patterns (see Appendix A). Therefore, these gaps form an air outlet passage between the degassing lines and the vacuum trench. Figure 2f shows a cross-sectional view of the fabricated disposable microchannel superstrate, which clearly shows the PET film, microchannel, degassing line, and degassing wall.

## 3. Results and Discussion

### 3.1. Degassing Test

The degassing capability of the PDMS degassing wall depends on its gas permeability and thickness and the experimental conditions, such as ambient pressure and temperature. The gas permeability of PDMS is related to the porosity of PDMS [44], which is determined by the curing temperature [45]. In this study, the PDMS prepolymer was cured at 75 °C to achieve high gas permeability [46]. In the lateral degassing method, the vacuum pressure used for the assembly and degassing ranges from −30 to −80 kPa. The solution and ambient temperature were kept at a room temperature of 25 °C. The degassing rate of the lateral degassing method was evaluated using a microfluidic device with closed meandering microchannel, as shown in Figure 3a. The degassing rate, *k*, can be calculated using the degassing flow rate, *dQ*/*dt*, per area, *A*, as
(1)$$k=-\frac{1}{A}\frac{dQ}{dt}$$
where *A* and *Q* are the degassing area of the degassing wall and the volume of air bubble, respectively, and *t* represents time. Assuming that the air bubble attached to the degassing wall as the shape of a quarter circle, the degassing area, *A*, and the volume, *Q*, of the bubble can be expressed as follows:(2)A=2 h r
(3)$$Q=\frac{\pi }{4}\;h\;r^{2}$$
where *r* is the radius of the bubble. Using Equations (2) and (3), Equation (1) can be transformed using the height of the microchannel, *h* (=50 μm) as follows:(4)$$\frac{dQ}{dt}=-\frac{4\sqrt{h}}{\sqrt{\pi }}k\sqrt{Q}$$
(5)$$Q={\left(\sqrt{Q_{0}}-\frac{2\sqrt{h}}{\sqrt{\pi }}kt\right)}^{2}$$
where *Q*_0_ is the initial volume of air bubble.

In the lateral degassing method, the degassing rate is mainly determined by the gas flow resistances of the PDMS degassing wall and the small gap between the PET film and the substrate. Therefore, it can be predicted that the thickness of the PDMS degassing wall and the surface roughness of the substrate are the main variables that determine the degassing rate. To measure the degassing rate, a liquid droplet was created in a Tygon tube (0.5 mm id) connected to the inlet, and the time taken by the liquid droplet to travel through the tube into the closed meandering microchannel, as well as the distance, were measured, as shown in Figure 3b. The thicknesses of the PDMS degassing walls were 30, 50, 70, 100, 150, and 200 μm. The data points in Figure 3c were obtained from three datasets, measured using a substrate with a 4.2 μm high micropillar array and a bare glass substrate. The measured results indicate that the degassing rate increases as the PDMS degassing wall becomes thinner. The data also imply that the degassing rate obtained using the substrate with the micropillar array increased slightly compared to that obtained using the bare glass substrate, because the gas can be easily evacuated through the micropillar array. In particular, when the height of the micropillar array exceeds 4.2 μm, it no longer acts as a gas flow resistance [47]. As the vacuum pressure increases and the thickness of the degassing wall becomes thinner, the difference in degassing rates between the two experiments gradually increases. In other words, in both cases, the flow resistance effect of the small gap between the PET film and the bare glass substrate becomes more pronounced as the amount of degassing increases. In experiments with the bare glass substrate, the degassing rate is approximately 20% lower than that obtained using the micropillar array substrate. However, the micropillar array is difficult to fabricate and can be easily damaged, thereby making it unsuitable for reuse and possibly limiting the functional implementation of the substrate. Therefore, in this study, although the degassing rate was slightly lower, the bare glass substrate was used for subsequent applications of the lateral degassing method.

On the other hand, due to the vacuum pressure to assemble the superstrate and the substrate, the microchannel would be slightly deformed laterally as the degassing line is on the side of the microchannel. Assuming the degassing wall thickness, *W*, as 100 μm, the maximum deflection, *δ_max_*, of the wall can be calculated by modeling it as a flat rectangular plate with two long edges fixed as follows:(6)$$\sigma_{max}=\frac{PLh^{4}}{384EI},\;\;I=\frac{LW^{3}}{12}$$
where *P* is the vacuum pressure (‒50 kPa) in the degassing line, *L* (>>100 μm) and *h* (50 μm) are the length and height of the degassing wall, respectively, *E* is the Young’s modulus of PDMS (750 kPa), and *I* is the moment of inertia of the degassing wall. According to Equation (6), the maximum deflection of the 100 μm thick degassing wall by the vacuum pressure is 13 nm, which means that the vacuum pressure has little effect on the microchannel deformation.

### 3.2. Bonding Stability 

To measure the maximum pressure in the microchannel that the PDMS degassing wall can withstand, a straight microchannel (Figure 4a) with a degassing wall was used and an air pressure of 0–850 kPa was applied to the microchannel through a pressure regulator. At this time, the disposable superstrate and the substrate were assembled using a vacuum pressure of −50 kPa, which was also applied to the degassing line to maintain a constant pressure for consistent burst testing. To ensure that the air pressure was applied evenly to the entire microchannel, two pressure sensors (XGZP6847001MPG; CFSensor, Wuhu, Anhui, China) were connected to each end of the microchannel, as shown in Figure 4a. As the applied pressure increased, the PDMS degassing wall was pushed toward the degassing line and eventually burst upon reaching the critical pressure (Figure 4b and Appendix A). For the burst test, the applied pressure was increased by 50 kPa every 5 min until the degassing wall burst, and the critical pressure was measured using three identical disposable superstrates (Figure 4c). The experimental results showed that the maximum internal pressure that the degassing wall can withstand is in the range of 650–850 kPa, regardless of the thickness of the degassing wall. Therefore, the maximum available internal pressure is 650 kPa, which is much higher than the 10–20 kPa required to drive the fluid flow in typical microfluidic devices.

### 3.3. Applications of Microfluidic Devices with Complex Structures

To demonstrate the usefulness of the lateral degassing method for complex structured microfluidic devices, meandering microchannels [48], microvortex devices [49], gradient micromixers [50], and herringbone micromixers [51] were fabricated by the conventional PDMS imprinting method with glass substrate and the disposable film-chip technique along with the lateral degassing method, respectively. Air bubbles are usually trapped in complex microstructures or at corners with a high surface-to-volume ratio. Many microfluidic devices have complex structures as key components within microchannels, and these complex structures can often lead to the formation of bubbles, which can degrade the performance. In the meandering microchannel, the air bubbles are often trapped at corners (Figure 5a), which changes the fluid resistance. The microvortex device has sudden expansion–contraction reservoirs within the microchannel, which can trap critically large-sized cells due to the shear gradient lift force in the expansion reservoirs. When the sample is filled for the first time, air bubbles are usually formed at the corners of the expansion reservoirs, as shown in Figure 5b. Because large-sized cells are hydrodynamically captured in the bay of the expansion reservoirs, bubbles trapped in the corners severely hinder this separation. In case of the gradient micromixer, a large amount of air is inevitably trapped in the middle microchannels due to non-synchronized liquid filling (Figure 5c), which reduces the function of the mixer. The herringbone mixer contains chevron-shaped herringbone grooves, which induces vortex flow in the microchannel. Because the herringbone grooves act like cavities within the microchannel, air bubbles are trapped (Figure 5d) and thereby the performance of the herringbone mixer is degraded. However, using the lateral degassing method, the trapped air bubbles disappeared within an average of 60 s, and the degassing time varied depending on the bubble size (Figure 6a–d). These results were compared with the experimental results (Figure 5a–d) of microfluidic devices, fabricated by the conventional PDMS imprinting method with glass substrate. In conclusion, the experimental results showed that the lateral degassing method is simple and effective in removing air bubbles from within the microchannels.

## 4. Conclusions

This study introduced the lateral degassing method which is compatible with the film-chip technique for using it in disposable microfluidic devices. Using the gas-permeable degassing wall and vacuum assembly of the disposable superstrate and reusable substrate, air bubbles inside the microchannel disappeared within 60 s. As expected, the degassing rate increased as the thickness of the degassing wall decreased and the vacuum pressure increased. At a vacuum pressure of −50 kPa, the degassing rate of the 30 μm thick degassing wall was found to be 3.2 × 10^−6^ m s^−1^. This implies that 1 nL air bubble formed inside the 50 μm high microchannel can be completely removed within 1 min, as described by Equation (5). This degassing rate is similar to the previously reported in-plane degassing methods [9,39] where a vacuum tube was directly connected to the degassing line. Previously developed degassing method that used the PDMS membrane [47] was difficult to handle due to the high flexibility of the PDMS membrane. A substrate with a micropillar array was required to support the PDMS membrane for gas emission and to prevent it from sticking to the substrate. However, the PET film used in the lateral degassing method was easy to handle, and its stiffness created a small gap with the substrate, thereby creating a degassing path without micro-supporters on the substrate. The burst test showed that the thinnest 30 μm degassing wall could withstand pressure inside the microchannel of 600 kPa or more, allowing the lateral degassing method to be used even in microfluidic applications that require high drive pressure. One of the advantages of the lateral degassing method is that it can be equipped with a variety of functions that generate energy fields, such as acoustic, electric, magnetic [43], and thermal fields [41] on the reusable substrate. Owing to the 12 μm thick ultrathin PET film and vacuum assembly, the microchannel of the disposable superstrate can be as close as 12 μm from the energy source of the reusable substrate, which allows the energy field to be effectively transferred into the microchannel. Therefore, features, such as simple implementation, high degassing rate, ease of use, high pressure resistance, and versatile applications, will make the lateral degassing method a key technique to address the chronic bubble formation problem that hinders the performance of microfluidic devices.

## Figures and Tables

**Figure 1 membranes-11-00316-f001:**
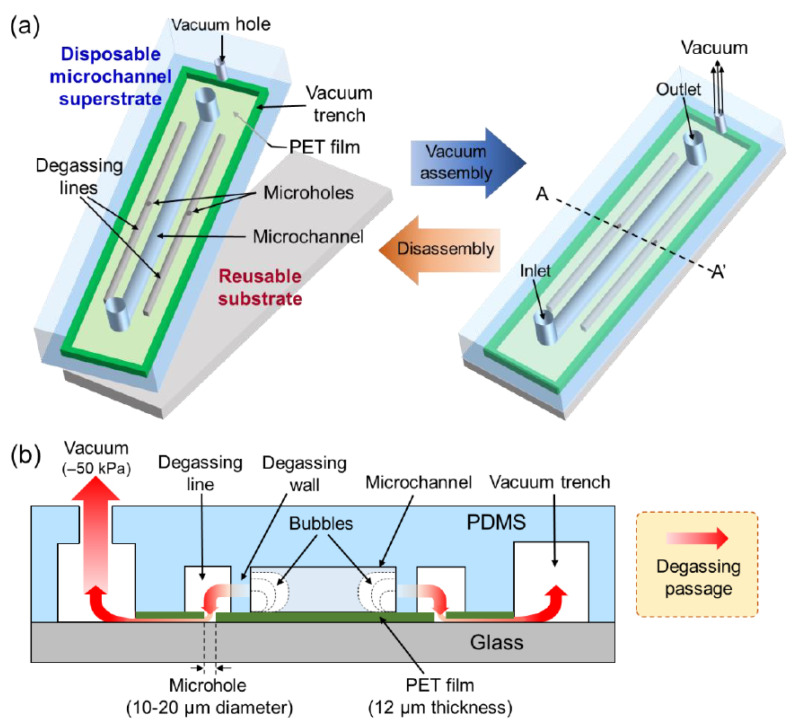
Working principle of the lateral degassing method. (**a**) Assembly and disassembly of the disposable microchannel superstrate and the reusable substrate by vacuum pressure. (**b**) Discharge of air bubbles.

**Figure 2 membranes-11-00316-f002:**
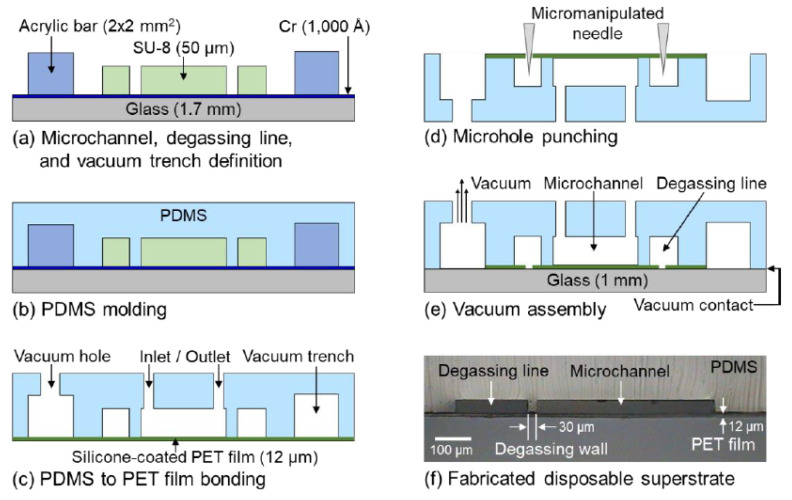
Fabrication process of the disposable microchannel superstrate with the lateral degassing method. (**a**) Cr deposition on a glass slide, followed by SU-8 patterning to define the microchannel and the degassing lines, and adhesive bonding of acrylic square bars to define the vacuum trench. (**b**) PDMS molding to create the microstructured PDMS replica. (**c**) Formation of inlet/outlet reservoirs and vacuum hole and bonding of the PDMS replica with the silicone-coated release PET film. (**d**) Creation of the microholes. (**e**) Vacuum assembly of the disposable microchannel superstrate and the glass substrate to produce the disposable film-chip microfluidic device. (**f**) A cross-sectional view of the fabricated disposable microchannel superstrate.

**Figure 3 membranes-11-00316-f003:**
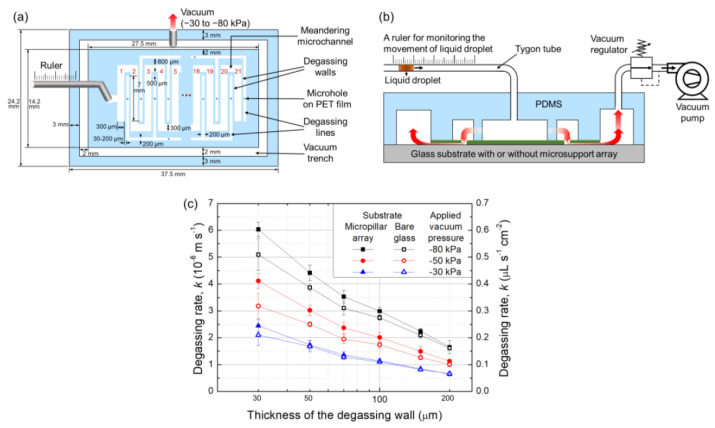
(**a**) Top view of the meandering microchannel for measuring the degassing rate using the lateral degassing method. A Tygon tube and a ruler were used to monitor the movement of a liquid droplet. (**b**) Experimental setup to measure the degassing rates according to the thickness of the degassing wall. (**c**) Degassing rates measured using a 4.2 μm high micropillar array substrate and a bare glass substrate.

**Figure 4 membranes-11-00316-f004:**
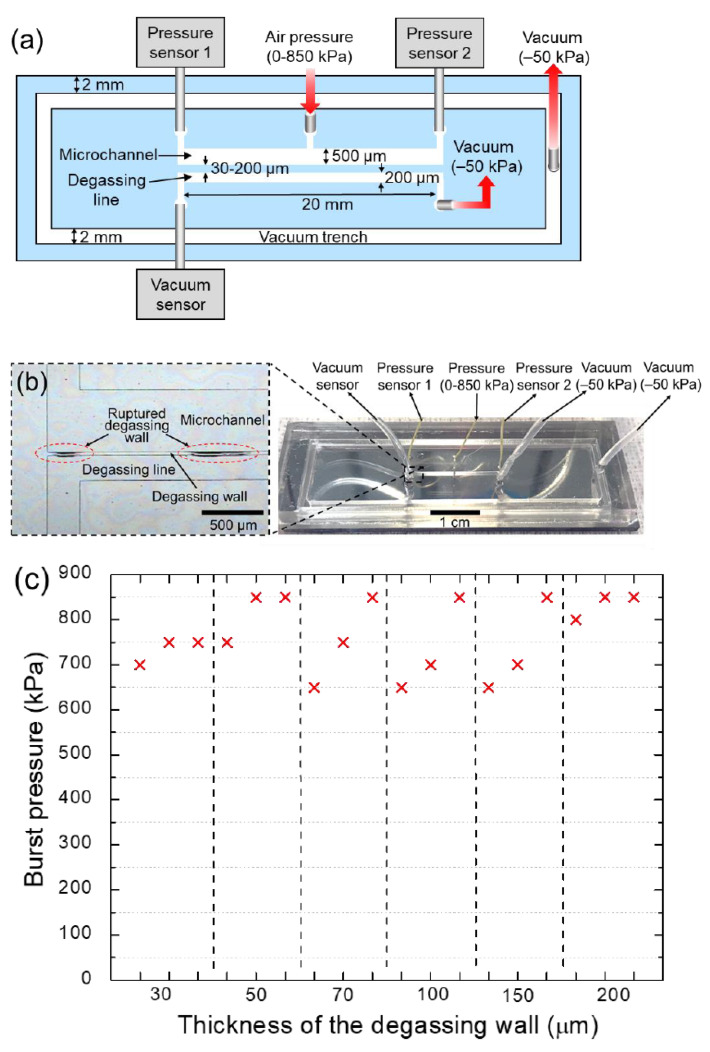
(**a**) Illustration of the instrumental setup for the burst test of the degassing wall. (**b**) Photograph of the disposable lateral degassing device for burst test of the degassing wall and enlarged view of the ruptured degassing wall. (**c**) Burst pressure measured according to the thickness of the degassing wall. The measurements were performed three times for each thickness of the degassing wall.

**Figure 5 membranes-11-00316-f005:**
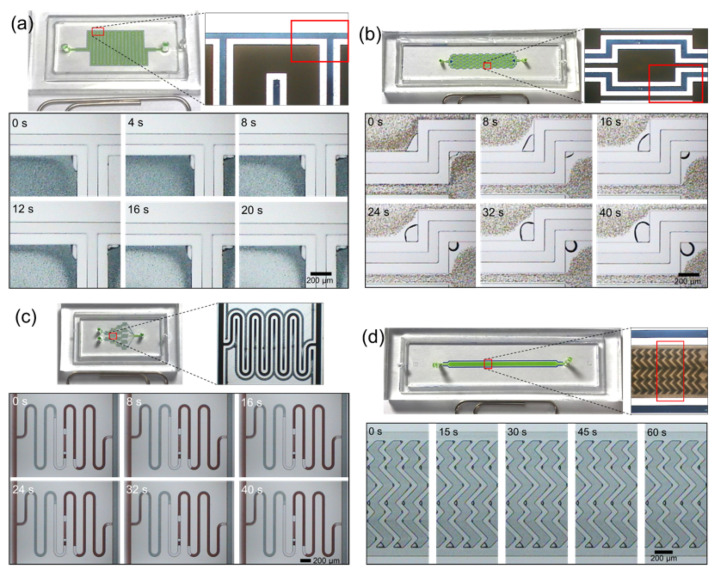
Sequential photographs showing air bubbles trapped in (**a**) a meandering microchannel, (**b**) a microvortex device, (**c**) a gradient micromixer and (**d**) a herringbone micromixer, fabricated by the conventional PDMS imprinting method using glass substrate.

**Figure 6 membranes-11-00316-f006:**
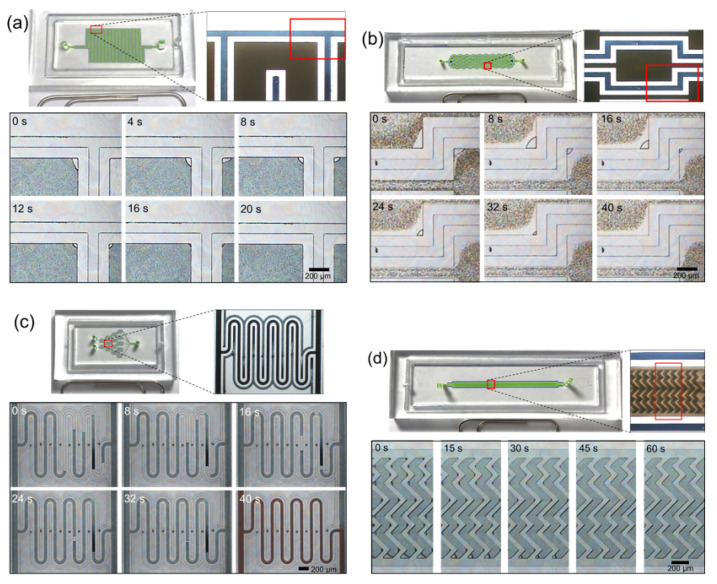
Sequential photographs showing bubble degassing in (**a**) a meandering microchannel, (**b**) a microvortex device, (**c**) a gradient micromixer and (**d**) a herringbone micromixer, fabricated by the disposable film-chip technique along with the lateral degassing method.

## Data Availability

Not applicable.

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
