# Peer review of "Lateral Degassing Method for Disposable Film-Chip Microfluidic Devices"

_membranes, 2021, doi:10.3390/membranes11050316_

Round 1
Reviewer 1 Report
Comments:
The manuscript intend to solve the problem for most of the microfluidic system, gas bubble stick on the channel wall. They used a lateral active degassing method that can be implemented in multi-layer microfluidic devices. The experimental result seems meaningful, and support their calm partly. However, there seem to be several serious issues with the manuscript. My detailed comments and possible approaches to improve the manuscript are mentioned below.
Major concerns:
- The reviewer always degas the microchannel by put it in a gas vacuum chamber before using it, and the PDMS will be in a lack of gas status, and will actively absorb the gas in the flow. Which is a quick and simple method. Authors claim this method is a quick and simple method for degassing, reviewer does not think so, since it requires extra components and multi-layer structure of device. Related content should be revised.
- What about the deformation of the main channel induced by degassing process. As you know PDMS is deformable and the change of the geometry induced failure on microfluidic application such as focusing [Cytometry Part A 97 (9), 909-920]. Some investigations with quantified data should be help.
- Apply negative pressure on PDMS channel indeed degassing the channel. But also absorb the gas in the media. For applications requiring precise culturing of biological sample, it is critical. A experimental investigation on this should be carried out.
- How the dimension of degassing channel influence on the effects of degassing should be indicated by using data. Not just wall thickness. And the dimensional of every channel in this manuscript should be indicated.
Other minor concerns must be addressed:
- The order of the fig.5 and 6 should be revised for better understanding, such as before and after.
- Scale bar in Fig.4(b) right image is missing
Author Response
Thank you for your valuable review. Please find an attached file for response to your comments.

Reviewer 2 Report
Degassing of microfluidic devices is a very interesting and practical topic, and a simple while effective method would be extremely helpful to researchers working in this field. The authors introduced a lateral degassing method using vacuum pressure to draw air from microchannel to degassing line in this manuscrip. They further studied the influence of degassing wall thickness on degassing rate, validated the robustness of the setup and demonstrated the application of this method with several complex microfluidic structures.
The manuscript is well-organized and easy to follow. The authors provided sufficient background references and detailed information in the methodology section. The experiments were reasonably designed and results were presented clearly. A few comments/questions:
- The authors stated that they developed a method which is simple to implement – how would they justify the difficulty of fabricating superstrates with additional degassing lines for complex microfluidic structures?
- When pressure is applied, how much will the PDMS superstrate deform and affect the transport of liquid?
- Line 173: it’s unclear how equation 2 was derived. What’s the correlation between A and h?
- The relevance of the work to membrane should be emphasized since the journal focuses on membrane-related research.
Author Response
Thank you for your valuable review. Please see the attachment for response to your comments.

Round 2
Reviewer 1 Report
The answer to the concern of deformation should be added in the main manuscript, it is critical to potential reader.
Author Response
Thank you for the review. Please see the attachment that responds to the comment.
